# Radiation-Induced Recurrent Vesicovaginal Fistula—Treatment with Adjuvant Platelet-Rich Plasma Injection and Martius Flap Placement—Case Report and Review of Literature

**DOI:** 10.3390/ijerph18094867

**Published:** 2021-05-03

**Authors:** Aleksandra Kołodyńska, Dominika Streit-Ciećkiewicz, Agata Kot, Iga Kuliniec, Konrad Futyma

**Affiliations:** 12nd Department of Gynecology, Medical University in Lublin, Jaczewskiego 8, 20-954 Lublin, Poland; dominika.streit@gmail.com (D.S.-C.); futymakonrad@mp.pl (K.F.); 2Healthcare Centre of St. John of God Independent Public Provincial Hospital in Lublin, Biernackiego 9, 20-400 Lublin, Poland; kotaga75@gmail.com; 3Hospice of the Good Samaritan, Bernardyńska 11a, 20-109 Lublin, Poland; 4Department of Urology and Oncological Urology, Medical University in Lublin, Jaczewskiego 8, 20-954 Lublin, Poland; iga.kuliniec@gmail.com

**Keywords:** vesicovaginal fistula, urogenital fistula, incontinence, radiation, Martius flap, platelet-rich plasma

## Abstract

Vesicovaginal fistula is the non-physiological connection between the urinary bladder and vagina. This results in continuous urine leakage. In developed countries, the prevalence of this condition is low and affects (mainly) women with a history of gynaecological procedures or radiotherapy. The aim of this study was to present the therapeutic process of a patient with radiation-induced, recurrent vesicovaginal fistula. The thirty-eight-year-old patient underwent radical hysterectomy with follow-up radiotherapy due to cervical cancer. Five years after the therapy, she was diagnosed with vesicovaginal fistula. After two unsuccessful Latzko procedures and two adjuvant platelet-rich plasma injections, a third Latzko reconstructive surgery was performed with additional transposition of the Martius flap—with successful closure of the fistula.

## 1. Introduction

Vesicovaginal fistula (VVF) is the non-physiological connection between the urinary bladder and vagina. This is a condition that results in continuous urine leakage. Compared to developing countries, where it is a common problem due to poor perinatal care, resulting in obstetrical fistulas, in developed countries, the prevalence of this condition is low and affects, for the most part, women with a history of gynaecological procedures or radiotherapy [1]. In the research of Härkki-Sirén et al., for example, a group of 62,379 hysterectomies was analysed, and the total incidence of vesicovaginal fistula was 0.8% [2]. According to Härkki-Sirén et al., laparoscopic hysterectomies had the highest risk of this complication, as compared with no incidence of VVF after supracervical abdominal hysterectomy.

Radiation-induced vesicovaginal fistulas, regardless of the size or localization, are classified as complex [3,4]. In 1997, Maier et al. analysed over 10,000 patients treated with radiotherapy due to gynaecological malignancies. Urinary fistula was the second most common urological complication, following irradiated bladder. The authors emphasized that although the risk of fistula formation is relatively low (0.3%), it can severely affect the patient’s condition [5] and quality of life.

Among patients with radiation-induced VVF, it is widely favoured to prevent the recurrence of the malignancy before they undergo therapeutic procedures [1].

## 2. Case Report

A thirty-eight-year-old women with a history of two spontaneous deliveries was diagnosed with planoepithelial cervical cancer IB-1, which was, according to FIGO, histologically G-3, in 2014 [6]. Following surgical treatment, she underwent a Meigs radical hysterectomy. Due to metastatic disease in the left iliac lymphatic nodes and advanced abdominal adhesions (resulting in right ureter occlusion), she qualified for percutaneous nephrostomy and radiotherapy. Ambulatory brachytherapy was applied—two applications on the upper vaginal cuff (1100 cGy/g). Afterwards, the patient developed postradiation symptoms, including mild dysuric symptoms, for a few days after the procedure. The follow-up treatment was teletherapy in the pelvis area with lymphatic system (4500/4500 cGy) involvement—with good tolerance. Moreover, hormone replacement therapy with transdermal oestrogens was introduced after radiotherapy.

In January 2015, due to increasing right hydronephrosis, psoas hitch and Boari flap ureterocystoneostomy and double J (DJ), catheter placement was performed. During the first 18 months postsurgery, the patient contracted several urinary tract infections (UTIs). Moreover, right-side hydronephrosis re-occurred, which resulted in the need for DJ catheter replacement and repeated antibiotic treatment. In July 2016, after another ineffective conservative treatment, the patient was scheduled for right nephrectomy.

In January 2018, the patient presented acute left kidney injury (AKI) with left postradiation ureter occlusion, and, therefore, underwent ureterorenoscopy (URS) with the DJ catheter left until December 2018. In May 2019, she was hospitalized due to a moderate depressive episode.

In July 2019, due to continuous urine leakage from the vagina, the patient was examined for the vaginal location of cancer recurrence. Speculum examination revealed a vesicovaginal fistula orifice in the vaginal rag on the right side (Figure 1). Computed tomography scan, cystoscopy and cystography were then performed, and no indication for additional histopathological examination was found, but the vesicovaginal fistula was confirmed.

In October 2019, the patient was admitted to our tertiary urogynaecological department, and Latzko vaginal reconstructive surgery was performed. A Foley catheter was kept in place for 18 days after the surgery. A week after catheter removal, the patient again presented a constant flow of urine from the vagina. A dye test with methylene blue indicated the recurrence of the fistula.

In December 2019, the patent qualified for a platelet-rich plasma (PRP) injection procedure of the fistula edges. This operation is described elsewhere [7]. In February 2020 (7 weeks after the injection), a 2nd Latzko reconstructive procedure was performed. The Foley catheter was kept in place this time for 17 days after surgery. A week after catheter removal, the patient again reported continuous urine leakage through the vagina, and the dye test revealed the recurrence of the fistula.

In May 2020, another PRP injection was performed. Five weeks later, a 3rd Latzko reconstructive surgery was performed with additional transposition of the Martius flap from the right labia majora (Figure 2). The Foley catheter was kept in place for 21 days after the surgery. Around the 10th day after the catheter removal, the patient observed periodic urinal leakage. The test with methylene blue was negative. A small amount of methylene blue leakage was, however, observed from the external urethral orifice, suggesting de novo stress urinary incontinence. Two months after the surgery with Martius flap transposition, the dye test (200 mL of methylene blue solution) was negative, and the cough test appeared to be positive. The wound was healing properly both in the area of the fistula and the labia majora (Figure 3). The patient was diagnosed with stress urinary incontinence.

In the 3-month follow up, while the patient complained of mild pain in her right labia majora, stress urinary incontinence symptoms had decreased, and the patient only required occasional urine pad use.

## 3. Discussion

In highly developed countries, the main aetiology of vesicovaginal fistula is iatrogenic trauma—either postoperative or following radiotherapy. This includes the vast majority of malignancy survivors. This condition can seriously affect the patient’s quality of life (QoL), both physically and psychosocially [8]. In a cohort study with 290 patients with VVF in Malawi, more than 90% of all patients reported a significant improvement in QoL after VVF successful closure. In the group of patients with persistent continuous urinary incontinence, 18% fulfilled the criteria for a likely depressive disorder, while 7% reported suicidal ideation [9]. Despite this, there are no clear guidelines concerning a VVF repair strategy, although the vaginal approach is believed to be the gold standard in the first surgical attempt.

Singh et al., in a prospective randomized study, compared repair surgeries of VVF with or without the adipose tissue flap interposition, both transvaginal and transabdominal routes. In the study, patients were randomised into four subgroups with or without flap interposition (either Martius flap or omental flap), and each of these had a control group. The success rate did not differ across the groups and was around 96%. The only difference in postoperative results was altered sensation or pain in the labia majora in the group with Martius flap interposition. However, the group included only patients with simple VVF [10].

In contrast, Rangnekar et al., in a group with vesicovaginal or urethrovaginal fistula caused by obstetric trauma, presented a 95.2% success rate among patients who underwent the Martius procedure, compared with 72% with anatomic repair. Surprisingly, dyspareunia appeared only in patients who underwent anatomic repair and was reported by 33.3%. Moreover, in the group with anatomic repair, 8.0% of all patients subsequently experienced stress urinary incontinence [11].

The application of minimally invasive techniques (laparoscopic or robotic) in surgical VVF repair can be observed to have significance [8]. The results are very promising, as they present a success rate similar to the open transabdominal technique, with proven benefits characteristic of minimally invasive routes (minor blood loss, time of hospitalization or analgesic demand) [12]. Unfortunately, radiation-induced fistulas are particularly very challenging for surgeons. The origin of this type of fistula is derived from fibrosis of the bladder wall’s lamina propria, with the presence of modified large fibroblasts called “radiation fibroblasts” and obliterative arteritis in the radiated area. As a result, the atrophy and necrosis of the bladder epithelium appears and leads to increased tension in the scarred, fibrous bladder and vaginal walls, followed by rupturing of the affected ulcerated area and evolution into fistulous canals [13,14].

All of the above processes that lead to the formation of the fistula can be possible factors that can disturb the healing process after the fistula operative management. It seems, therefore, rational to delay the repair of the fistula for about 1 year after the radiation in order to allow the surrounding tissues to become histologically stable again [14].

Pushkar et al. analysed 210 patients with radiation-induced VVF, who either underwent transvaginal repair surgery using the traditional splitting repair, Latzko colpocleisis or Martius flap approaches. The success rate after primary repair was only 48.1%, but after follow-up attempts, the cumulative success rate grew, respectively, by 66.6%, 77.1% and 80.4% after more than 3 surgeries. The authors emphasized that subsequent repairs do not decrease the chance for the patient to be cured. On the contrary, patients should be carefully informed that the total closure of the fistula might require several attempts. The most probable cause of this fact is continuing histological rearrangements of the tissue after the radiation [14].

It is commonly believed that research should focus on evaluating the main predictors that give rise to accurately premeditating patients to the most appropriate mode of surgical treatment leading to the successful closure of the fistula in the first attempt—with the patient informed about the prognosis.

Barone et al. conducted a multicenter study to identify the factors of adverse outcome of VVF repair (such as the failure of closing the fistula or residue urinary incontinence). There were three groups of factors analysed in the study: the patient’s demographics and medical history; characteristic of the fistula; and other factors related directly to the surgery, i.e., the surgeon’s experience, materials used and technical approach. The overall success rate among the patients included to the study was 81.7%, but nearly 20% of the patients with closed fistula reported residual incontinence. The authors identified several risk factors for fistula closure failure. These included small bladder size, prior repair, severe vaginal scarring (based on the surgeon’s subjective assessment) and involvement of the urethra. The risk of residual incontinence among patients with successful closure of the VVF was also associated with prior repair, vaginal scarring (1.3 times greater risk) and urethral involvement (2 times greater risk). It was interesting that none of the contextual factors, such as the surgeon’s experience, had influence on the therapy’s success [15].

Bengston et al. developed a scoring system so as to recognize patients with a high risk of residual urinary incontinence after VVF surgical repair. The results showed that factors such as age, number of years with fistula, prior surgery at an outside facility, fistula size, circumferential fistula, vaginal scarring, bladder size and urethral length significantly influence the risk of postoperational residual urinary incontinence. The calculated cut-off point in the developed model showed sensitivity and specificity at 82% and 63%, respectively. Thus, the risk score model of Benston et al. allows the identification of patients who may benefit from repairs performed by more experienced surgeons or that involve other additional procedures [16].

To increase the effectiveness of VVF, especially with a high risk of failure to close the fistula or residual incontinence, multiple studies have been undertaken on supporting therapies that could enhance the effects of the surgeries. Blood-based products, such as autologous platelet-rich plasma and fibrin glue, seem to be the most promising, and this is confirmed by the growing number of clinical cases and enhanced research.

Fibrin glue was first used in surgical procedures in 1974 as a haemostatic agent and tissue adhesive [17]. It contains fibrinogen and thrombin in a concentration that is higher than that used in physiologic procedures; thus, it supports haemostasis, enables the formation of a coagulum, prevents fibrosis and amplifies collagen synthesis [18]. Another advantage of fibrin glue is its complete biodegradability within weeks after the application. The use of fibrin glue in the treatment of urological conditions, including one case of vesicovaginal fistula, was described by Sharma S. et al. The authors reported the successful closure of a fistula after the endoscopic injection of a fibrin glue with no following complications [19].

Platelet-rich plasma (PRP) is a small volume of autologous plasma with highly concentrated platelets. As a potential source of growth factors, i.e., angiogenic and miogenic, as well as adhesive and chemotactic proteins, its properties are widely used in a number of medical specialties [20]. The use of PRP started under mostly non-operative conditions, such as healing promotion after musculoskeletal injuries.

In the field of gynaecology, PRP therapy is a promising, non-invasive therapeutic modality in multiple conditions, including wound healing, vulvar dystrophy and premature ovarian failure, as well as in urogenital conditions, such as vesicovaginal fistulas [21].

The preliminary results of the research of Streit-Ciećkiewicz et al. on patients with recurrent vesicovaginal fistulas showed the successful surgical closure of a fistula 4–6 weeks after PRP injection with no complications or adverse reactions [7]. In this research, an attempt was made to find factors that might have had predictive value when assigning the patient surgical treatment. Surprisingly, it was found that low BMI should be considered as a single risk factor of surgical failure, and patients should be assigned auxiliary adipose tissue transposition in the first attempt [4].

Interestingly, it was shown in another study that women with significantly higher BMIs were more likely to succeed with the fistula closure after the first surgery [4]. Such outcome could be caused by higher fat content and fatty tissue histological properties. It was found that a subpopulation of adipose tissue stem cells, called stromal vascular fraction cells (SVF cells), as well as preadipocytes, fibroblasts, leukocytes, macrophages and endothelial cells, play important roles in modulating and improving the properties of healing tissues. Moreover, among the most relevant auxiliary surgical methods used in urogenital fistulae closure is the Martius flap technique. This is based on the interposition of labial fatty tissue with all its components, mentioned above, but also the preservation of vessels supplying blood and enhancing the rapid neovascularization of the surrounding tissues; it also prevents the overlapping of the urinary and vaginal wall suture lines [4,11,22,23,24].

Further studies concerning the predictive factors of surgical success of VVF are required.

## 4. Conclusions

Vesicovaginal fistulas have an undeniable impact on a patient’s quality of life and still pose a great challenge for urogynaecological surgeons. The literature concerning the route or technique of the surgery is not unanimous, especially in the case of radiation-induced VVF. The identification of risk factors for recurrence and exploring new supportive therapies is essential for improving the diagnostic process and treatment. Additionally, the combined influence of PRP and adipose tissue on improving surgical wound healing should be investigated, especially in a context in which platelets should be injected in order to achieve the maximal effect.

## Figures and Tables

**Figure 1 ijerph-18-04867-f001:**
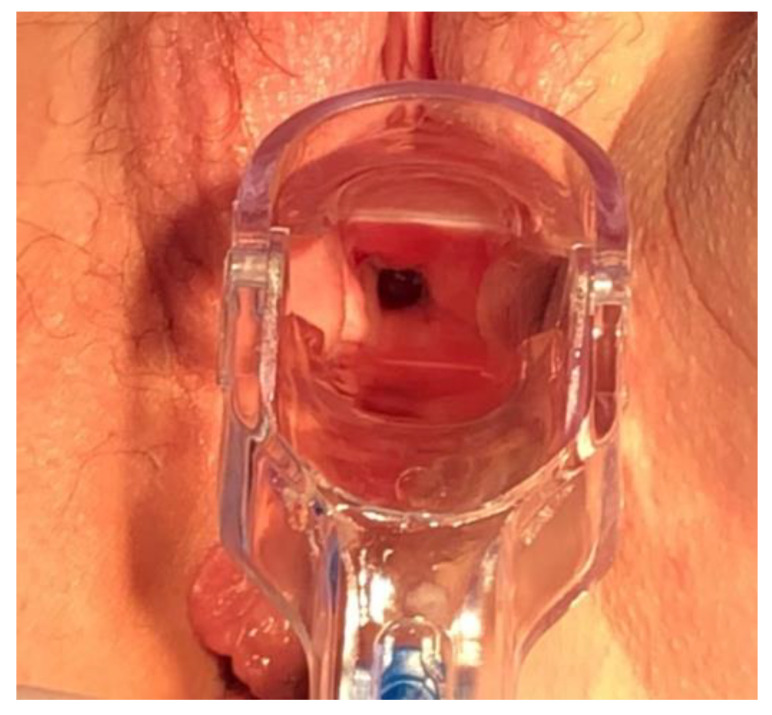
Vaginal orifice of the fistula.

**Figure 2 ijerph-18-04867-f002:**
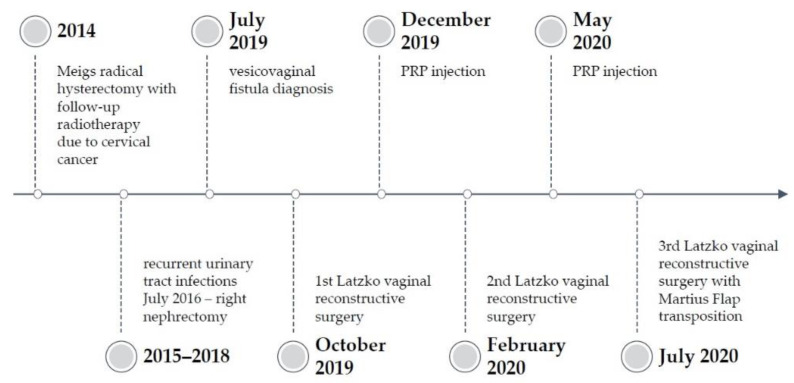
Timeline of the procedures.

**Figure 3 ijerph-18-04867-f003:**
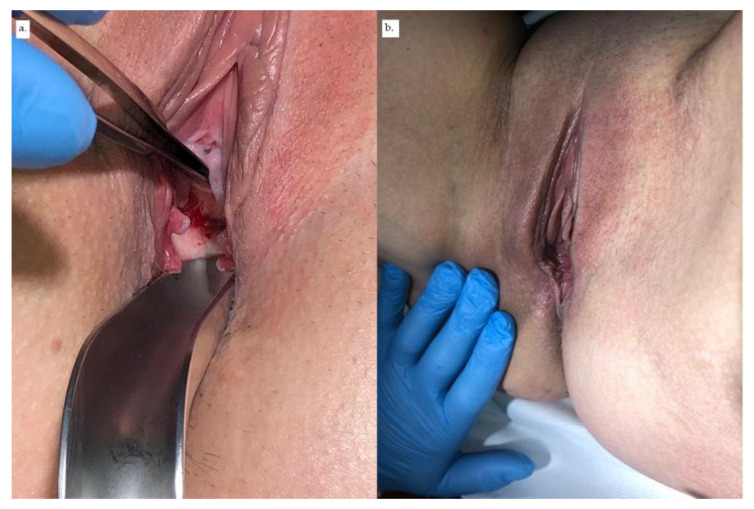
(**a**) Wound healing of the area of fistula vaginal opening. (**b**) Healed area of the right labia majora.

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
