# Peer review of "Radiation-Induced Recurrent Vesicovaginal Fistula—Treatment with Adjuvant Platelet-Rich Plasma Injection and Martius Flap Placement—Case Report and Review of Literature"

_ijerph, 2021, doi:10.3390/ijerph18094867_

Round 1

Reviewer 1 Report

Although the incidence of VVF is extremely low, it will seriously affect women's quality of life. In this case report, the author describes clearly the process of the VVF case in detail and proposes feasible treatments to solve the repeatedly plagued health problems. Such experience sharing and literature review and discussion can indeed provide more effective treatment strategies for the development of VVF therapy.

Suggestion:

  1. The commas marked with numbers on lines 35 and 40 are missing.
  2. Author may visualize the possible important determinants that affect the health recovery of this case, and discuss or compare with the possible influencing factors proposed in the discussion references, so as to provide specific reference for the future treatment of VVF.

Author Response

Although the incidence of VVF is extremely low, it will seriously affect women's quality of life. In this case report, the author describes clearly the process of the VVF case in detail and proposes feasible treatments to solve the repeatedly plagued health problems. Such experience sharing and literature review and discussion can indeed provide more effective treatment strategies for the development of VVF therapy.

Thank you for your opinion.

Suggestion:

  1. The commas marked with numbers on lines 35 and 40 are missing.

We are sorry for the mistakes in the text editing. Thank you for your careful analysis. We added commas marked with numbers on lines 35 and 40.

  1. Author may visualize the possible important determinants that affect the health recovery of this case, and discuss or compare with the possible influencing factors proposed in the discussion references, so as to provide specific reference for the future treatment of VVF.

We have added a short paragraph describing possible mode of PRP and adipose tissue mode of action: “Interestingly, as it was shown in another study that women with significantly higher BMI were more likely to succeed with the fistula closure after the first surgery [4]. Such outcome could be caused by the higher fat content and fatty tissue histological properties. It was found that a subpopulation of adipose tissue stem cells, called stromal vascular fraction cells (SVF cells), as well as preadipocytes, fibroblasts, leukocytes, macrophages and endothelial cells play important roles in modulating and improving healing tissues’ properties. Moreover, one of the most relevant auxiliary surgical methods used in urogen-ital fistulae closure is the Martius flap technique. It is based on interposition of labial fatty tissue with all its components, mentioned above, but also with preserving vessels sup-plying blood and enhancing rapid neovascularization of the surrounding tissues, it also prevents overlapping of the urinary and vaginal walls suture lines [4, 11, 22-24].” – lines 216-227.

Reviewer 2 Report

Dear Author,

I read with interest the manuscript entitled “Radiation-induced recurrent vesicovaginal fistula - treatment 2 with adjuvant platelet-rich plasma injection and Martius flap 3 placement – case report and review of literature”.

This case study illustrates a typical clinical history of patients with vesicovaginal fistula. The review of the literature appears comprehensive, highlighting the psychological impact of such issue and the necessity of expert surgical team management.

The entire manuscript is well written.

My only comment would be to precise, page 2, that the Latzko procedure is a vaginal approach for readers not used to that technic.

Many thanks for letting me review this manuscript.

Yours sincerely,

Author Response

Dear Author,

I read with interest the manuscript entitled “Radiation-induced recurrent vesicovaginal fistula - treatment 2 with adjuvant platelet-rich plasma injection and Martius flap 3 placement – case report and review of literature”.

This case study illustrates a typical clinical history of patients with vesicovaginal fistula. The review of the literature appears comprehensive, highlighting the psychological impact of such issue and the necessity of expert surgical team management.

The entire manuscript is well written.

My only comment would be to precise, page 2, that the Latzko procedure is a vaginal approach for readers not used to that technic.

Many thanks for letting me review this manuscript.

Yours sincerely,

Thank you for your careful revision of our study. We ensure the information about the Latzko procedure in text (line 80).

Reviewer 3 Report

The authors reported a single patient with vesicovaginal fistula. Then, they presented literature review on the etiology and treatment of VVF. This manuscript is very interesting. But, in my opinion, the manuscript is not suitable for publication in the journal of IJERPH, because the main observation it described was reported one case and literature review.

Author Response

The authors reported a single patient with vesicovaginal fistula. Then, they presented literature review on the etiology and treatment of VVF. This manuscript is very interesting. But, in my opinion, the manuscript is not suitable for publication in the journal of IJERPH, because the main observation it described was reported one case and literature review.

Thank you for your review. In developed countries, recurrent VVFs are not a very common condition, but they seriously influence the patient’s quality of life. In our opinion, the description of the case together with the discussion may contribute to the development of effective therapies. This case report was submitted to “Frontiers in Urogynecology-Changing Perspectives on LUTS Pathophysiology and Treatment” IJERPH special issue and in our opinion, perfectly fits into the subjects of new modes of treatment difficult cases.

Reviewer 4 Report

Dear Authors,

The manuscript deals with an extremely important topic concerning complications after anti-cancer therapy, directly affecting the quality of life of women. The described case seems very interesting, however the description is very concise, with plain text only, describing a lot of procedure the patient undergone. My suggestion is to elaborate more on the procedures (and its effect on subject) and add some graphs/tables/photos to visualise the described case and the timeline of procedures.

The abbreviations should be explained in text (VVF, DJ).

The conclusions should be more specific and concern not only the discussed literature, but most of all the discussed case. Please rephrase.

Author Response

Dear Authors,

The manuscript deals with an extremely important topic concerning complications after anti-cancer therapy, directly affecting the quality of life of women. The described case seems very interesting, however the description is very concise, with plain text only, describing a lot of procedure the patient undergone. My suggestion is to elaborate more on the procedures (and its effect on subject) and add some graphs/tables/photos to visualise the described case and the timeline of procedures.

Thank you for your opinion. We added photos and a timeline graph to the manuscript (Figure 1 – line 76-77, line 72 in text, Figure 2 – line 101-102, line 92 in text, Figure 3 – line 104-106, line 98-99 in text).

The abbreviations should be explained in the text (VVF, DJ).

Thank you for your careful analysis. We explained the abbreviations in the text (lines 29 and 61).

The conclusions should be more specific and concern not only the discussed literature but most of all the discussed case. Please rephrase.

We rewrote the conclusions (lines 231 – 238).

Round 2

Reviewer 4 Report

Dear Authors,

thank you for your response and including suggested changes in manuscript. 

At this point I have no further suggestions.